# AN AGNOSTIC APPROACH TO FEDERATED LEARNING WITH CLASS IMBALANCE

**Zebang Shen, Juan Cervino, Hamed Hassani, Alejandro Ribeiro**
Department of Electrical and Systems Engineering
University of Pennsylvania
Philadelphia, PA 19104, USA
`{zebang,jcervino,hassani,aribeiro}@seas.upenn.edu`

## ABSTRACT

Federated Learning (FL) has emerged as the tool of choice for training deep models over heterogeneous and decentralized datasets. As a reflection of the experiences from different clients, severe class imbalance issues are observed in real-world FL problems. Moreover, there exists a drastic mismatch between the imbalances from the local and global perspectives, i.e. a local majority class can be the minority of the population. Additionally, the privacy requirement of FL poses an extra challenge, as one should handle class imbalance without identifying the minority class. In this paper we propose a novel agnostic constrained learning formulation to tackle the class imbalance problem in FL without requiring further information beyond the standard FL objective. A meta algorithm, CLIMB, is designed to solve the target optimization problem, with its convergence property analyzed under certain oracle assumptions. Through an extensive empirical study over various data heterogeneity and class imbalance configurations, we showcase that CLIMB considerably improves the performance in the minority class without compromising the overall accuracy of the classifier, which significantly outperforms previous arts. In fact, we observe the greatest performance boost in the most difficult scenario where every client only holds data from one class. The code can be found here.

## 1 INTRODUCTION

Class imbalance is ubiquitous in real world supervised learning problems. Examples span from medical applications (Lee & Shin, 2020; Roy et al., 2019; Choudhury et al., 2019), fraud detection (Yang et al., 2019; Chan et al., 1999), to consumer based applications (Wang et al., 2021b; Wu et al., 2020; Long et al., 2020). In these scenarios, data belonging to a subset of classes constitute a great proportion of the population while data from the minority classes, generated by uncommon events, are scarce (He & Garcia, 2009). Having a non-uniform number of samples per class deteriorates the performance of the classifier in the minority class (Huang et al., 2016), resulting in low training and testing accuracy. More importantly, unintended consequences of mistreating the minority class can be catastrophic (Van Hulse et al., 2007) if the problem is not handled appropriately.

From the perspectives of *data heterogeneity* and *privacy*, the issue of class imbalance is even more significant in the setting of Federated Learning (FL). Due to the heterogeneity of the local data distributions, there can be a significant mismatch between the local and global imbalance, i.e. the class that is a minority locally can actually be a majority class globally. Moreover, for the purpose of privacy protection in FL, one should tackle the class-imbalance problem in an agnostic manner, i.e. the proposed algorithms *should not* require the minority class to be identified.

In the centralized training setting, techniques like balanced sampling, loss re-weighting, and gradient tuning have achieved many successes (Johnson & Khoshgoftaar, 2019). However, due to the extra difficulty of handling the class imbalance problem in FL, previous arts that rely on the explicit identification of the minority class do not directly apply. While research along this line is quite limited, a commonly used heuristic is to estimate the portion of a class using the norm of the gradient per class. When used as proxies, these quantities are then utilized to reweight the losses

corresponding to different classes (Wang et al., 2021a; Yang et al., 2020). A key drawback of (Wang et al., 2021a) is that a subset of the client's local dataset needs to be shared, which is not ideal for privacy preserving purposes. Moreover, the effectiveness of such approaches degrades when there is a notable mismatch between the local imbalance and the global imbalance (Wang et al., 2021a).

In this work, we target the most difficult yet possibly most interesting setting in FL, where there is a significant mismatch between the local and global imbalance. Concretely, on certain clients, a minority class from a global perspective is in the majority locally. Practitioners often encounters such a mismatch due to the highly heterogeneous data configuration in FL and the *principle of locality* (Wang et al., 2021a). For example, when the local data are produced by a minority user, the corresponding minority data could occupy a large portion of the local distribution. In this scenario, due to the scarceness of the minority data from a global perspective, the corresponding underrepresented client will usually experience poor service quality from the trained model, giving rise to potentially severe fairness issue.

To overcome the aforementioned challenges, we propose a constrained learning formulation to handle the class imbalance issue in FL while accounting for both *heterogeneity* and *privacy*. In brief, we impose constraints on the standard FL formulation so that the empirical loss on every client should not overly exceed the average empirical loss. Such constraints are shown to force the classifier to account for all classes equally and hence mitigate the detrimental effect of class imbalance, under a type of heterogeneous data configuration that captures the mismatch between the local and global imbalance. The advantages of our formulation are threefold: First, in contrast to previous arts which usually rely on some heuristics to reweight the loss functions for different classses, our approach is principled, yielding a simple optimization interpretation. Second, unlike existing methods, our formulation requires no additional information compared to the original FL formulation and it treats data from different classes agnostically, and hence is less likely to leak the private information of the clients. Third, from our extensive empirical study, our formulation can significantly improve the testing accuracy on the minority class, without compromising the overall performance.

**Contribution.** We summarize our contributions as follows.

1. **Problem Formulation:** To address the challenge of class imbalance in FL with data heterogeneity, we propose a novel constrained FL formulation with explicit enforcement on the similarity between the local empirical losses. Using only information from the standard FL objective, our approach is completely agnostic to class distribution of the client data and its proxies, as opposed to the existing literature (which mostly violate privacy requirements of FL).

2. **Meta-Algorithm:** We solve our constrained optimization problem via a primal-dual approach. For a fixed dual variable, the corresponding Lagrangian function enjoys a similar structure as the standard FL loss, but with non-uniform weights on the local objectives. Accordingly, we propose an efficient (meta-) algorithm called `CLIMB` that can use any FL optimization method as a subroutine, with nearly negligible communication overhead. Furthermore, we analyze the convergence properties of `CLIMB` under certain oracle assumptions.

3. **Accuracy Improvement in the minority class:** On benchmark datasets, `CLIMB` with Fed-Avg as base solver achieves significant enhancement of the accuracy in the minority class without compromising the overall accuracy under various data heterogeneity and class imbalance scenarios.

## 2 RELATED WORKS

**Class Imbalance in Centralized Setting.** In the centralized learning setting, the number of samples per class is known. We categorize existing solutions that exploit such information as follows.

*Balanced sampling.* In order to balance the data used for gradient calculation, the most common approaches are either to under sample from the majority classes (Liu et al., 2008), or to augment the minority class (Chawla et al., 2002; Guo & Viktor, 2004). Both methods seek to generate an artificial uniform distribution of data (Buda et al., 2018; Pouyanfar et al., 2018).

*Loss reweighting.* Methods in this category focus on re-weighting the loss functions for different classes, with extra emphases on the mistakes made in the minority class (Cui et al., 2019; Ling & Sheng, 2008; Sun et al., 2007; Wang et al., 2016). There also exist works like (Lin et al., 2017) that adjust the scale of the loss sample-wise, without exploiting the global data distribution.

*Gradient tuning.* In the context of neural networks, some other works focus on tuning the gradient per class (Anand et al., 1993) showing faster rates of convergence in the class imbalance setting.

To do this, the gradient is reshaped in order to have an equal magnitude when projected to all the gradients per class, while keeping the same norm as the original gradient.

**Class Imbalance in Federated Learning.** Combating with the issue of class imbalance is more challenging in FL since the data composition in such a setting is generally unknown and the minority classes are often difficult to identify Li et al. (2020); Wang et al. (2020b). While research in this direction are quite limited, to the best of our knowledge, we classify them into the following two classes based on whether a proxy of the data composition is explicitly built.

**(i)** A common strategy in existing methods is to explicitly build proxies of the data composition based on some heuristics. These proxies are dynamically adjusted during the learning procedure and allow the aforementioned three techniques, balanced sampling, loss re-weighting and gradient tuning, to be utilized in the FL setting: Wang et al. (2021a) suggests that there exists a proportional relation between the magnitude of the gradients the corresponds to the last layer of the neural network and the sample quantity. Based on such observation, the proposed the `Ratio-Loss`, a class-wise re-weighted version of the standard cross entropy loss. We note that to define the `Ratio-Loss`, one needs to collect a subset of client data as "auxiliary data", which may not be ideal in the setting of FL. On the same vein, Yang et al. (2020) argues that the gradient per class can be used as a proxy to infer the imbalances in the distributions of the clients, and thus clients should be selected according to how uniform the magnitude of gradient per class is. We also notice the work `Astraea` which introduces extra virtual components called mediators between the FL server and clients (Duan et al., 2019). The mediator is assumed to have access to the local data distributions of the clients so that client rescheduling and data augmentation can be carried out accordingly. Through the prepossessing on the mediator, the gradients communicated to the server are made balanced class-wise. However, when privacy is taken into consideration, methods like `Astraea` may not be appropriate as in essence it is directly built on the global data distribution.

**(ii)** Without explicitly constructing an estimation of the data composition, there are works that resort to techniques like active learning and reinforcement learning to implicitly learn the data composition as the optimization progresses. Works along this research line usually rely on client selection to mitigate the effects of class imbalances. In (Goetz et al., 2019), to address the class imbalance problem, there will a higher probability to sampling clients that posses the global minority class. Other works, leverage client selection as a multi-armed bandit problem, and design a client selection policy to balance the gradients (Xia et al., 2020). Other arts leverage Q-learning techniques to select clients in order to minimize the overall loss Wang et al. (2020a). However, we believe in the most practical setting of FL, clients that are available per communication round is incoercible and hence these strategies have limited applicability.

**Comparison with Previous Works.** Our method significantly differs from the above strategies in the following ways: (1) Our method requires no knowledge of the data composition, nor any proxy of such information. Consequently, our approach is agnostic to the minority class and better preserves clients' privacy. Such a property draws clear distinction between our work and the works listed in class (i) above. (2) As will be more clear in the following section, we only perform client-wise re-weighting as opposed to the class-wise re-weighting schemes used in previous works, which further emphasize the agnostic nature of our approach. (3) In contrast to the methods in class (ii) above, our approach does not require active client selection, enjoying a broader applicability in the most practical and interesting settings.

## 3 FEDERATED LEARNING WITH EMPIRICAL LOSS CONSTRAINTS

We consider the problem of multi-class classification. Let $x \in \mathcal{X} \subseteq \mathbb{R}^d$ be the input and use $y \in \mathcal{Y} = \{1, \ldots, C\}$ to denote the target label, where $C$ is the total number of classes. In the context of FL, we assume to have $N$ clients, each of which posses its own private data distribution $p_i(x, y)$. Here, $p_i(x, y)$ is a joint probability distribution of the input $x$ and the output label $y$. One can decompose $p_i(x, y)$ as $p_i(x, y) = p(x|y)p_i(y)$, where $p(x|y)$ is the conditional distribution of the input $x$ given class $y$ and $p_i(y)$ is the marginal distribution of class $y$ on client $i$. We assume that the conditional distribution $p(x|y)$ is identical on all devices, but the marginal distribution $p_i(y)$ can vary significantly due to the heterogeneity of the data configuration.

Let $\mathcal{H} = \{\phi : \mathcal{X} \times \Theta \to \mathbb{R}^C\}$ be a family of parameterized predictors with parameter $\theta \in \Theta \subseteq \mathbb{R}^Q$. Let $\ell : \mathbb{R}^C \times \mathcal{Y} \to \mathbb{R}_+$ be the loss function, then the local objective function of client $i$ is defined as

$$f_i(\theta) = \mathbb{E}_{(x,y)\sim p_i}[\ell(\phi(x, \theta), y)]. \tag{1}$$

For multi-class classification, $\ell$ is often chosen to be the cross entropy loss. In the standard formulation of FL, the goal is to minimize the global average of local objectives

$$\min_{\theta \in \Theta} \bar{f}(\theta) := \frac{1}{N} \sum_{i=1}^{N} f_i(\theta). \tag{2}$$

While it is well known that, in the presence of class imbalance, the above vanilla formulation will produce models that perform poorly on the minority data, we consider the following configuration of heterogeneous local class distributions in order to make a quantitative analysis of such a phenomenon for a concrete class imbalance setting. More importantly, such a setting will also motivate our constrained FL formulation. We emphasize that the following setting is just used as a motivating example to showcase our claims, and our results apply generally to any FL setting.

**Motivating Example.** Let $u$ be the uniform distribution over the classes, i.e. for $y \sim u$, $\Pr(y = c) = \frac{1}{C}, \forall c \in \mathcal{Y}$ and let $\delta_c$ be the Dirac distribution of class $c$. We assume for some small but fixed $\alpha \in [0, 1]$, the local class distribution of the client $i$ is a mixture of the uniform distribution over all classes and the Dirac distribution of a fixed class $c_i$, i.e. $p_i = \alpha u + (1 - \alpha)\delta_{c_i}$. We use $N_c$ to denote the number clients with $c_i = c$. Note that in the limit setting when $\alpha = 0$, the aforementioned configuration captures the most heterogeneous setting: clients only have data from a single class.

We consider an extreme case of class imbalance under the above configuration. Without loss of generality, we consider the binary classification problem, i.e. $C = 2$, and we assume class 1 to be the minority class with $N_1 = 1$. We define $g_i(\theta) := \mathbb{E}_{x \sim p(x|y=i)}[\ell(\phi(x, \theta), i)]$ as the loss of the predictor $\phi(\cdot, \theta)$ on the data with $y = i$. We can calculate that

$$\bar{f}(\theta) = \left(\frac{\alpha}{2} + \frac{1-\alpha}{N}\right) g_1(\theta) + \left(\frac{\alpha}{2} + \frac{(1-\alpha)(N-1)}{N}\right) g_2(\theta). \tag{3}$$

Clearly, when $\alpha$, the portion of data with uniform label distribution, is small, e.g. $\alpha = 0$, and $N$, the total number of clients, is large, the loss of the predictor on class 1 has negligible weight, which often leads to the poor performance on the minority class in the trained classifier as the majority classes dominate the gradient. Moreover, observe that when $\alpha$ is small, there is a significant mismatch between the local and global class imbalance: the global minority class 1 is in the majority on the corresponding client locally. Such phenomenon is pertinent to the FL setting due to the data heterogeneity and poses a great challenge for tackling the issue of class imbalance in FL.

## 3.1 Constrained FL Formulation

In our work, to address the class imbalance challenge, we propose to minimize the following constrained FL (CFL) formulation

$$P_\epsilon^* = \min_{\theta \in \Theta} \bar{f}(\theta) := \frac{1}{N} \sum_{i=1}^{N} f_i(\theta) \tag{CFL}$$

$$\text{s.t. } f_i(\theta) - \bar{f}(\theta) \leq \epsilon, \forall i \in [1, \ldots, N].$$

Here, $\epsilon$ is a tolerance constant that controls the enforced closeness in the training loss among clients, and we emphasize the dependence of the optimal value $P_\epsilon^*$ on $\epsilon$ by encoding it in the subscript. As a motivation, we show that the constraint in our formulation (CFL) can be translated to a constraint on the performance of the minority class, under the above class imbalance setting: Consider the setting where the tolerance constant is very small, i.e. $\epsilon$ is close to zero. WLOG, we assume that the first device is the one that has $c_i = 1$. Recall the definition of $g_i$ above Eq.(3). One can compute that

$$f_1(\theta) - \bar{f}(\theta) = \frac{(1-\alpha)(N-1)}{N}(g_1(\theta) - g_2(\theta)) \leq \epsilon \iff g_1(\theta) - g_2(\theta) \leq \frac{N\epsilon}{(1-\alpha)(N-1)}.$$

Therefore, our formulation (CFL) enjoys a clear class-balancing interpretation in the highly heterogeneous setting when $\alpha$ is small and $N$ is large. Note that this is achieved *without* the identification of the minority class and server collects *no* additional information compared to the vanilla FL.

While the above nice interpretation of the proposed constraint may not hold exactly for general class imbalance settings, in spirit, we want the resulting classifier from our algorithm to perform similarly on every class, or in other words to account for the minority class and majority class equally. In the following section, we discuss how to solve the proposed formulation by alternating the primal and dual updates of an equivalent Lagrangian formulation.

---

**Algorithm 1** CLIMB: CLass IMBalance Federated Learning

---

1: **Input**: initial model $\theta^0$, a subroutine *ClientUpdate*, dual step size $\eta_D$, maximum round $T$;
2: Initialize the dual variables $\boldsymbol{\lambda} = [0, \ldots, 0]$;
3: **for** $t = 1, 2, \ldots, T$ **do**
4:     Compute weights: $\forall i \in [N], w_i = 1 + \lambda_i - \bar{\lambda}$, with $\bar{\lambda} = \frac{1}{N} \sum_{i=1}^N \lambda_i$;
5:     **Primal Update**: $\theta^{t+1} \leftarrow ClientUpdate(\{w_i\}_{i=1}^N, \theta^t)$;
6:     **Dual Update**: $\forall i \in [N], \lambda_i \leftarrow [\lambda_i + \eta_D(f_i(\theta^{t+1}) - \bar{f}(\theta^{t+1}) - \epsilon)]_+$, with $\bar{f} = \frac{1}{N} \sum_{i=1}^N f_i$;
7: **end for**
8: **Output**: model $\theta^{T+1}$

---

## 3.2 ALGORITHM CONSTRUCTION

In order to solve problem (CFL), we resort to the method of Lagrange multipliers. By introducing the dual variables $\boldsymbol{\lambda} = [\lambda_1, \ldots, \lambda_N] \in \mathbb{R}_+^N$, we define the Lagrangian function as,

$$\mathcal{L}(\theta, \boldsymbol{\lambda}) = \frac{1}{N} \sum_{i=1}^N f_i(\theta) + \lambda_i \left( f_i(\theta) - \frac{1}{N} \sum_{j=1}^N f_j(\theta) - \epsilon \right) \tag{4}$$

$$= \frac{1}{N} \sum_{i=1}^N (1 + \lambda_i - \bar{\lambda}) f_i(\theta) - \lambda_i \epsilon, \qquad \text{with } \bar{\lambda} = \frac{1}{N} \sum_{i=1}^N \lambda_i. \tag{5}$$

With this in hand, we can construct a lower bound of the above constrained optimization problem using the Lagrangian $\mathcal{L}(\theta, \boldsymbol{\lambda})$ as follows:

$$D_\epsilon^* = \max_{\boldsymbol{\lambda} \in \mathbb{R}_+^N} \min_{\theta \in \Theta} \mathcal{L}(\theta, \boldsymbol{\lambda}) \leq \min_{\theta \in \Theta} \max_{\boldsymbol{\lambda} \in \mathbb{R}_+^N} \mathcal{L}(\theta, \boldsymbol{\lambda}) = P_\epsilon^*,$$

where we often refer to $D_\epsilon^*$ as the dual problem. To solve the dual problem, we propose CLIMB, a method described in Algorithm 1, which is discussed in detail as follows.

CLIMB proceeds by alternatingly optimizing over the primal variable $\theta$ and the dual variable $\boldsymbol{\lambda}$.
**Primal Update.** For a fixed $\boldsymbol{\lambda}$, the minimization of $\mathcal{L}$ with respect to $\theta$ is equivalent to a re-weighted version of the standard unconstrained FL objective (2):

$$\min_{\theta \in \Theta} \mathcal{L}(\theta, \boldsymbol{\lambda}) \iff \min_{\theta \in \Theta} \frac{1}{N} \sum_{i=1}^N (1 + \lambda_i - \bar{\lambda}) f_i(\theta). \tag{6}$$

Importantly, this simple and canonical form allows us to perform the update on $\theta$ using any FL solver as the base optimizer. For flexibility, we do not explicitly choose the base optimizer in our algorithm description, but refer to the update on $\theta$ as the subroutine $ClientUpdate(\{w_i\}_{i=1}^N, \theta^t) \rightarrow \theta^{t+1}$. Such a subroutine takes the non-uniform weights $\{w_i\}_{i=1}^N$ on the local objectives and the current consensus model $\theta^t$ as inputs, and returns an updated model $\theta^{t+1}$. Note that every call to *ClientUpdate* may consists of multiple communication rounds.
**Dual Update.** Once we have obtained the new consensus model $\theta^{t+1}$, we perform dual update on $\lambda$ by taking a *single* dual ascent step of the following equivalent objective (given some fixed $\theta$),

$$\max_{\boldsymbol{\lambda} \in \mathbb{R}_+^N} \mathcal{L}(\theta, \boldsymbol{\lambda}) \iff \max_{\boldsymbol{\lambda} \in \mathbb{R}_+^N} \frac{1}{N} \sum_{i=1}^N \lambda_i \left( f_i(\theta) - \frac{1}{N} \sum_{j=1}^N f_j(\theta) - \epsilon \right). \tag{7}$$

To evaluate the gradient of $\mathcal{L}$ with respect to $\boldsymbol{\lambda}$, we need to first broadcast the consensus model $\theta^{t+1}$ and then aggregate the function values $f(\theta^{t+1})$. Since the broadcast model can be used for the next round of primal update, the only overhead of CLIMB compared to the standard FL solver *ClientUpdate*, is to transmit the functional value, which is negligible.

**Remark 3.1** *We emphasize that CLIMB can be implemented in a privacy-preserving manner: A client can carry out its update locally given the access to the global average of the dual variables $\bar{\lambda}$ and the global average of the loss functions $\bar{f}(\theta)$. These quantities can be computed via the standard FL technique of Homomorphic Encryption without revealing the exact value of the dual variable $\lambda_i$ and local loss $f_i(\theta)$ to the server, as elaborated in Appendix D.*

### 3.3 THEORETICAL GUARANTEES

Based on the recent progress in constrained learning (Chamon et al., 2021), we show that under mild regularity conditions, the duality gap between the dual problem $D_\epsilon^*$ and the primal problem $P_\epsilon^*$ can be controlled by some quantity that describes the capability of the parametric function class $\mathcal{H}$.

**Assumption 3.1** *The loss function $\ell$ in the definition of the local objective (1) is L-Lipschitz, i.e.$\|\ell(x,\cdot) - \ell(z,\cdot)\| \le L\|x - z\|$, and bounded by $B$.*

**Assumption 3.2** *The conditional distribution $p(x|y)$ is non-atomic for all $y \in \mathbb{R}^C$.*

While usually the local data distribution is discrete, we can always augment it by randomly perturbing the data points with white noise, this is often used as data augmentation in vision tasks.

**Assumption 3.3** *There exists a convex hypothesis class $\hat{\mathcal{H}}$ such that $\mathcal{H} \subseteq \hat{\mathcal{H}}$, and there exists a constant $\xi > 0$ such that $\forall \hat{\phi} \in \hat{\mathcal{H}}$, there exists $\theta \in \Theta$ such that $\sup_{x \in \mathcal{X}} \|\hat{\phi}(x) - \phi(x,\theta)\| \le \xi$.*

A simple strategy to construct $\hat{\mathcal{H}}$ is to take the convex hull of $\mathcal{H}$. When $\mathcal{H}$ is sufficiently rich, $\xi$ can be expected to be small. Notice that this bound can be decreased by increasing the richness of the function class $\mathcal{H}$. All the proofs can be found in the Appendix.

**Theorem 3.1 (Near Zero Duality Gap)** *Under Assumptions 3.1, 3.2,3.3, and the Contrained Federated Learning problem is feasible in $\hat{\mathcal{H}}$ with constraint $\epsilon - 2L\xi$, the Constrained Federated Learning problem has near zero-duality gap,*

$$P_\epsilon^* - D_\epsilon^* \le (2|\boldsymbol{\lambda}_{\epsilon - 2L\xi}^*|_1 + 1)L\xi, \tag{8}$$

*where $\boldsymbol{\lambda}_{\epsilon - 2L\xi}^*$ is the optimal dual variable associated with the Constrained Federated Learning problem (CFL) with constraints $\epsilon - 2L\xi$ over the space of functions $\hat{\mathcal{H}}$.*

Theorem 3.1 establishes an upper bound on the duality gap of the Constrained Federated Learning Problem CFL. The gap depends on the Lipschitz constant of the loss function $L$, the richness of the function class $\xi$, and the optimal dual variable of a more restrictive problem. Note that we required the Constrained Federated Learning problem to be feasible for constraint $\epsilon - 2L\xi$. In the case of the cross-entropy loss, as long as $\epsilon - 2L\xi > 0$, this can be satisfied by a classifier that assigns a uniform label for every sample, as each individual loss will be equal to each other.

Since the minimization of the Lagrangian function $\mathcal{L}$ is non-convex, to show the convergence of CLIMB, we need an additional oracle assumption as follows.

**Assumption 3.4 (Approximate Solution)** *For every dual variable $\boldsymbol{\lambda} \in \mathbb{R}_+^N$, and precision $\delta > 0$ there exists an oracle approximate solution $\theta_{\boldsymbol{\lambda}}$ such that $\mathcal{L}(\theta_{\boldsymbol{\lambda}}, \boldsymbol{\lambda}) \le \min_{\theta \in \mathbb{R}^Q} \mathcal{L}(\theta, \boldsymbol{\lambda}) + \delta$.*

**Theorem 3.2 (Convergence)** *Define the dual function $d(\boldsymbol{\lambda}) = \min_{\theta \in \Theta} \mathcal{L}(\theta, \boldsymbol{\lambda})$. Under Assumptions 3.1 to 3.4, for a fixed tolerance $r > 0$, the iterates generated by Algorithm 1 converge to a neighborhood of the dual problem $D_\epsilon^*$ in at most $T_r = \mathcal{O}(1/r)$ steps, i.e.,*

$$d(\boldsymbol{\lambda}^{T_r}) \ge D_\epsilon^* - \delta - \frac{\eta_D}{2}B^2 - r, \tag{9}$$

## 4 EXPERIMENTS

In this section, we evaluate our formulation (CFL) against the competitors on various FL scenarios at the presence of class imbalance. Our results highlight the benefits of our approach especially when the local data distributions are severely heterogeneous and there is a significant mismatch between local and global imbalance. To ensure a fair comparison, we use Fed-Avg as the base optimizer in the current experiment for *all formulations*: the standard FL objective in Eq.(2), the proposed formulation in Eq.(CFL), Ratio-Loss (Wang et al., 2021a), and Focal-Loss (Lin et al., 2017). We emphasize that the novelty of our work lies in the new constrained FL formulation (CFL) and is orthogonal to how the formulation is solved. We now describe the datasets and models used in our experiments with more details provided in Appendix A.

**Datasets** Three benchmark datasets are used in our experiments with the default train/test splits,

| Imbalance ratio | Dataset | Level of heterogeneity | Baseline (Eq.(2)) | CLIMB (this work) | Ratio-Loss | Focal-Loss |
|---|---|---|---|---|---|---|
| $\rho = 20$ | CIFAR10 | | 1 minority class out of 10 total classes | | | |
| | | $\alpha = 0.1$ | 0.0532 (0.6754) | **0.2080** (**0.6829**) | 0.1140 (0.6727) | 0.0450 (0.6445) |
| | | $\alpha = 0.2$ | 0.137 (0.7121) | **0.3230** (**0.7121**) | 0.1790 (0.7037) | 0.0430 (0.6914) |
| | | | 3 minority classes out of 10 total classes | | | |
| | | $\alpha = 0.1$ | 0 (0.5669) | **0.2810** (**0.6031**) | 0 (0.5746) | 0 (0.6565) |
| | | $\alpha = 0.2$ | 0.1279 (0.7098) | **0.3240** (**0.7167**) | 0.1790 (0.7054) | 0.0552 (0.6905) |
| | MNIST | | 1 minority class out of 10 total classes | | | |
| | | $\alpha = 0.1$ | 0.6889 (0.9375) | **0.8552** (**0.9556**) | 0.8472 (0.9544) | 0.6186 (0.9278) |
| | | $\alpha = 0.2$ | 0.7925 (0.9540) | **0.8748** (**0.9616**) | 0.8052 (0.9555) | 0.7784 (0.9479) |
| | | | 3 minority classes out of 10 total classes | | | |
| | | $\alpha = 0.1$ | 0.3425 (0.8260) | **0.6987** (**0.9158**) | 0.4134 (0.8484) | 0.1944 (0.7938) |
| | | $\alpha = 0.2$ | 0.4720 (0.8596) | **0.7290** (**0.9153**) | 0.6717 (0.9063) | 0.4602 (0.8654) |
| $\rho = 10$ | CIFAR10 | | 1 minority class out of 10 total classes | | | |
| | | $\alpha = 0.1$ | 0.2058 (0.6841) | **0.3629** (**0.7041**) | 0.2164 (0.6839) | 0.0414 (0.6543) |
| | | $\alpha = 0.2$ | 0.1813 (0.7113) | **0.3743** (**0.7312**) | 0.2657 (0.7083) | 0.1347 (0.6911) |
| | | | 3 minority classes out of 10 total classes | | | |
| | | $\alpha = 0.1$ | 0.0492 (0.5933) | **0.2280** (**0.6358**) | 0.0315 (0.5825) | 0 (0.5548) |
| | | $\alpha = 0.2$ | 0.1064 (0.6380) | **0.2734** (**0.6696**) | 0.0993 (0.6211) | 0.0298 (0.5982) |
| | MNIST | | 1 minority class out of 10 total classes | | | |
| | | $\alpha = 0.1$ | 0.8473 (0.9534) | **0.9305** (**0.9584**) | 0.8851 (0.9558) | 0.8469 (0.9470) |
| | | $\alpha = 0.2$ | 0.8962 (0.9634) | **0.9239** (**0.9657**) | 0.8798 (0.9615) | 0.8953 (0.9586) |
| | | | 3 minority classes out of 10 total classes | | | |
| | | $\alpha = 0.1$ | 0.6045 (0.8906) | **0.8084** (**0.9356**) | 0.6981 (0.9119) | 0.4690 (0.8670) |
| | | $\alpha = 0.2$ | 0.7272 (0.9190) | **0.8195** (**0.9392**) | 0.7663 (0.9320) | 0.6961 (0.9099) |

Table 1: The minority class testing accuracy and the overall testing accuracy (the quantity in the parentheses) after 5000 communication rounds. If there are multiple minority classes, we report the worst of them. Here $N$, the number of devices, is 500. The base FL solver is Fed-Avg with *partial-participation*: 100 devices participate in every communication round.

which are MNIST (LeCun et al., 1998), CIFAR10 (Krizhevsky et al., 2009) and Fashion-MNIST (Xiao et al., 2017). The results on the last dataset are deferred to the appendix due to space limitation. *Heterogeneity.* We generate heterogeneity in the local data distributions according to the strategy from (Karimireddy et al., 2020; Hsu et al., 2019): Let $\alpha \in [0, 1]$ be some constant that determines the level of heterogeneity. For a fixed $\alpha$, we divide the dataset among $N = 100$ (moderate) or $N = 500$ (massive) clients as follows: for we allocate to each client a portion of $\alpha$ i.i.d. data and the remaining portion of $(1 - \alpha)$ by sorting according to label. In our appendix, we also consider the Dirichlet type heterogeneous data allocation scheme which is wildly used in the literature of Federated Learning, for example (Hsu et al., 2019; Acar et al., 2020).
*Data Imbalance.* We simulate the phenomenon of class imbalance by removing data belong to the minority classes: Observe that the datasets included in our experiments both have 10 perfectly balanced classes. For the minority class(es), we retain only $1/\rho$ portion of the corresponding data. Here, $\rho \geq 1$ the ratio between the numbers of data in the majority class and in the minority class

and is termed the *imbalance ratio*. For example, when there are 3 minority classes with $\rho = 10$, $90\%$ of the data belong to classes 0, 1, 2 (without loss of generality) are manually removed. In our experiments, we consider the setting of 1 or 3 minority classes and we take $\rho = 5, 10, 20$.

**Models** We follow the choice of model architectures in (Acar et al., 2020; McMahan et al., 2017). Specifically, we use a 2 hidden layer fully-connected neural network for MNIST, where the numbers of neurons are (128, 128). For CIFAR10, we use a CNN model consisting of 2 convolutional layers with 64 $5 \times 5$ filters followed by 2 fully connected layers with 394 and 192 neurons. We note that higher testing accuracy on the included datasets can be obtain by using models with high capacity, but is orthogonal to our research.

## 4.1 RESULTS SUMMARY

To evaluate the effectiveness of an approach against the challenge of class imbalance, one needs to take into consideration both the performance on the minority class(es) and the average performance on all the classes. We report both quantities after sufficient communication rounds (5000 rounds for CIFAR10 and 1000 rounds for MNIST) under various experiment settings in Tables 1 and 2. In every cell of these tables, the quantity above denote the minority class testing accuracy and the quantity in the parentheses is the average performance on all the classes. We also considered the case where there are multiple minority classes and we report the worst accuracy among the minority classes. Our approach outperforms previous arts in all cases, often by a large margin.

**Imbalance Ratio and Number of Minority Classes.** The imbalance ratio $\rho$ and the number of minority classes are two important quantities to measure the difficulty of a class imbalance problem. Under all the included choices, CLIMB consistently beats previous arts in both minority class testing accuracy and average performance on all the classes. Therefore, we conclude that CLIMB is able to boost the performance on the minority class *without* compromising the performance on the other classes. This is a rare merit when addressing the class imbalance problem. In fact, methods like Ratio-Loss is able to improve the testing accuracy on the minority class, but it also sacrifices the performance on the rest classes, leading to an inferior average testing accuracy.

**Level of Heterogeneity.** We test the performance of CLIMB under different levels of heterogeneity and observe that CLIMB outperforms the included methods considerably in all settings. It has the biggest advantage over the competitors in the most heterogeneous setting, $\alpha = 0$. There are situations that existing methods completely fail in the minority class with less than $10\%$ accuracy, but CLIMB is still able to correctly classify most of the minority data, e.g. see Table 2$\to \rho = 5 \to$ MNIST$\to$ 3 minority classes$\to \alpha = 0$.

**Moderate vs. Large Number of Devices** An important goal of FL is to exploit the computational resources of the IoT devices. Hence, the scalability to a large number of devices is a critical property of an FL method. We hence test CLIMB on both moderate ($N = 100$, see Table 2) and massive devices ($N = 500$, see Table 1) settings. To make things more practical, we instantiate the subroutine *ClientUpdate* using Fed-Avg with the partial-participation scheme in the massive device setting. Specifically, 100 devices participate model training after every Fed-Avg global communication round. We clearly observe the advantage of CLIMB in all of these setups.

## CONCLUSION

In this paper we proposed a novel agnostic constrained learning formulation to tackle the problem of class imbalance in the Federated Learning setting. By introducing constraints in the learning procedure we enforced the performance to be similar in all clients, thus accounting for the class imbalances. In terms of privacy protection, our formulation requires no further information than the standard FL objective to be collected in the server and it never estimates the data composition as opposed to all previous approaches. Moreover, compared with previous arts which are usually heuristic based, our approach is principled as it is purely optimization based and can be efficiently solved via the proposed meta-algorithm CLIMB, yielding major practical benefits. Our extensive empirical study showcases the superiority of proposed constrained formulation over previous arts.

## ACKNOWLEDGMENTS

This research is supported by AFOSR Award 19RT0726, NSF HDR TRIPODS award 1934876, NSF award CPS-1837253, NSF award CIF-1910056, NSF CAREER award CIF-1943064, and NSF award CCF-2112665.

| Imbalance ratio | Dataset | Level of heterogeneity | Baseline (Eq.(2)) | CLIMB (this work) | Ratio-Loss | Focal-Loss |
|---|---|---|---|---|---|---|
| $\rho = 10$ | CIFAR10 | | 1 minority class out of 10 total classes | | | |
| | | $\alpha = 0.0$ | 0.0229 (0.5734) | **0.5575** **(0.6076)** | 0 (0.4836) | 0 (0.4205) |
| | | $\alpha = 0.1$ | 0.2753 (0.7143) | **0.5054** **(0.7246)** | 0.2929 (0.6951) | 0.2284 (0.6860) |
| | | $\alpha = 0.2$ | 0.2988 0.7348 | **0.4689** **(0.7511)** | 0.3825 (0.7329) | 0.2618 0.7249 |
| | | | 3 minority classes out of 10 total classes | | | |
| | | $\alpha = 0.0$ | 0.0402 (0.5534) | **0.2756** **(0.5598)** | 0 (0.4678) | 0 (0.4527) |
| | | $\alpha = 0.1$ | 0.1316 (0.6189) | **0.3399** **(0.6637)** | 0.0690 (0.615) | 0.0408 (0.5976) |
| | | $\alpha = 0.2$ | 0.2566 (0.6659) | **0.3292** **(0.6983)** | 0.1916 (0.6504) | 0.1213 (0.6346) |
| | MNIST | | 1 minority class out of 10 total classes | | | |
| | | $\alpha = 0.0$ | 0.3092 (0.8630) | **0.9175** **(0.9341)** | 0.4650 (0.8630) | 0.3078 (0.8556) |
| | | $\alpha = 0.1$ | 0.8597 (0.9586) | **0.9428** **(0.9675)** | 0.9189 (0.9648) | 0.8348 (0.9529) |
| | | $\alpha = 0.2$ | 0.8750 (0.9640) | **0.9377** **(0.9715)** | 0.9283 (0.9706) | 0.8882 (0.9631) |
| | | | 3 minority classes out of 10 total classes | | | |
| | | $\alpha = 0.0$ | 0.0153 (0.7133) | **0.8029** **(0.9115)** | 0.0631 0.7222 | 0.0717 0.7401 |
| | | $\alpha = 0.1$ | 0.7263 (0.9189) | **0.8828** **(0.9522)** | 0.7870 (0.9341) | 0.6674 (0.9069) |
| | | $\alpha = 0.2$ | 0.7950 (0.9352) | **0.8004** **(0.9416)** | 0.7801 (0.9364) | 0.7785 (0.9274) |
| $\rho = 5$ | CIFAR10 | | 1 minority class out of 10 total classes | | | |
| | | $\alpha = 0.0$ | 0.2892 (0.6382) | **0.5987** **(0.6468)** | 0.0942 (0.5506) | 0.1491 (0.5631) |
| | | $\alpha = 0.1$ | 0.4101 (0.7186) | **0.6075** **(0.7351)** | 0.4011 (0.7008) | 0.3558 (0.6994) |
| | | $\alpha = 0.2$ | 0.5335 (0.7536) | **0.6063** **(0.7556)** | 0.5054 (0.7427) | 0.4359 (0.7380) |
| | | | 3 minority classes out of 10 total classes | | | |
| | | $\alpha = 0.0$ | 0.0313 (0.4742) | **0.3813** **(0.5639)** | 0.0512 (0.5108) | 0 (0.4514) |
| | | $\alpha = 0.1$ | 0.5135 (0.6636) | **0.6420** **(0.6977)** | 0.4600 (0.6632) | 0.4213 (0.6384) |
| | | $\alpha = 0.2$ | 0.5251 (0.6960) | **0.6328** **(0.7202)** | 0.5751 (0.6883) | 0.5311 (0.6749) |
| | MNIST | | 1 minority class out of 10 total classes | | | |
| | | $\alpha = 0.0$ | 0.7245 (0.8953) | **0.9154** **(0.9224)** | 0.8020 (0.9016) | 0.7429 (0.8992) |
| | | $\alpha = 0.1$ | 0.9378 (0.9666) | **0.9582** **(0.9693)** | 0.9286 (0.9651) | 0.9408 (0.9624) |
| | | $\alpha = 0.2$ | 0.9448 (0.9711) | **0.9670** **(0.9734)** | 0.9561 (0.9733) | 0.9481 (0.9686) |
| | | | 3 minority classes out of 10 total classes | | | |
| | | $\alpha = 0.0$ | 0.0160 (0.7722) | **0.8744** **(0.9137)** | 0 (0.7370) | 0.0544 (0.7826) |
| | | $\alpha = 0.1$ | 0.8294 (0.9422) | **0.9217** **(0.9606)** | 0.8520 (0.9501) | 0.7987 (0.9361) |
| | | $\alpha = 0.2$ | 0.8557 (0.9536) | **0.8818** **(0.9595)** | 0.8730 (0.9536) | 0.8321 (0.9442) |

Table 2: The minority class testing accuracy and the overall testing accuracy (the quantity in the parentheses) after sufficiently many communication rounds. If there are multiple minority classes, we report the worst of them. Here $N$, the number of devices, is 100. The base FL solver is Fed-Avg with *full-participation*: all devices participate in every communication round.

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
