# OpenReview forum: "An Agnostic Approach to Federated Learning with Class Imbalance"
_ICLR.cc/2022/Conference — ICLR 2022 Poster_

### Official Review · Reviewer_Mp3G · 2021-11-02

**Correctness:** 3
**Technical Novelty And Significance:** 3
**Empirical Novelty And Significance:** 3
**Recommendation:** 6
**Confidence:** 3

**Main Review:**

Strengths:
1. The problem is clearly stated and the algorithm is well-organised, theoretical analysis is provided;
2. The problem itself is interesting, non-iid and imbalance is a key challenge in federated learning and real-world setting;
3. The proposed algorithm is simple but efficient, since it does need sharing local data, privacy of local clients is also kept. Also, no prior knowledge required matches the real-world setting;
4. The way of partitioning the data is reasonable;
5. The experiments are mostly sound and comprehensive. Code is provided in the material, thus it should be easy to repeat the experiments and utilize the algorithm;

Weaknesses:
1. More citations on non-iid/imbalanced federated learning should be cited or taken into consideration as baselines. For example, [1][2][3];
2. Some minor written issues need to be taken care of. For example, g(\theta) needs to be defined before using it;
3. In the paper, the authors stated that "in the extreme case where every device has only one class, these methods have inferior performance than the classic Fed-Avg method," it would be good if the authors can provide the results in appendix. Also, in some non-extreme settings, if some of those algorithms perform better than FedAvg, it should be considered as a baseline;
4. The experiments are good, but would be better if results of experiments on other datasets, for example, synthetic dataset, fashion-MNIST are provided.



[1] Tackling the objective inconsistency problem in heterogeneous federated optimization. J Wang, Q Liu, H Liang, G Joshi, HV Poor - arXiv preprint arXiv:2007.07481, 2020

[2] H. Wang, Z. Kaplan, D. Niu and B. Li, "Optimizing Federated Learning on Non-IID Data with Reinforcement Learning," IEEE INFOCOM 2020 - IEEE Conference on Computer Communications, 2020, pp. 1698-1707, doi: 10.1109/INFOCOM41043.2020.9155494.

[3] On the convergence of fedavg on non-iid data. X Li, K Huang, W Yang, S Wang, Z Zhang - arXiv preprint arXiv:1907.02189, 2019

**Summary Of The Paper:**

This paper focuses on an interesting and challenging setting in federated learning, where number of classes and samples are imbalanced across different local workers. Compared to other methods which might require subset of local data or prior knowledge of the minor classes, the algorithm proposed in this paper does not need neither local data nor prior knowledge, which maintains the privacy principle in federated learning and aligned to real-world settings. The main contributions are:
1. Proposes a new algorithm solving the non-iid issue in federated learning which only adds constraints on local losses, without sacrificing local data privacy;
2. Utilizes a primal-dual optimization mechanism to solve the problem, provides theoretical guarantees on convergence;
3. Based on the experiments, there is a significant performance increase compared to FedAvg.

**Summary Of The Review:**

This paper proposes a new algorithm on imbalanced setting in federated learning. It enjoys the advantages of not leaking local data and no need of prior knowledge and overall the algorithm based on primal-dual optimization is easy to understand and be utilized. It is a good paper after minor revisions.

---

> ### Author Response · Authors · 2021-11-23
> **Response to Reviewer Mp3G**
>
> We thank the reviewer for the detailed and well explained feedback. We agree with the points that the reviewer made and we have implemented the modifications to address them.
>
> > More citations on non-iid/imbalanced federated learning should be cited or taken into consideration as baselines.
>
> We have included the citations mentioned by the reviewer. We thank the reviewer for pointing these important references.
>
> > Some minor written issues need to be taken care of. For example, g(\theta) needs to be defined before using it.
>
> We have corrected the text so as to define the function before using it.
>
> > In the paper, the authors stated that "in the extreme case where every device has only one class, these methods have inferior performance than the classic Fed-Avg method," it would be good if the authors can provide the results in appendix. Also, in some non-extreme settings, if some of those algorithms perform better than FedAvg, it should be considered as a baseline
>
> We agree with the reviewer that we should provide the evidence that FedPD has inferior performance than Fed-Avg for highly heterogeneous setting (i.e. small $\alpha$) and the results are reported in Figure 3 of our revised submission. Due to time limitation, we only report the performance of FedPD under two heterogeneity settings. In our future revision, we will add more comparison with FedPD and test the performance of FedPD under non-extreme settings.
>
> > The experiments are good, but would be better if results of experiments on other datasets, for example, synthetic dataset, fashion-MNIST are provided.
>
> We thank the reviewer for the positive feedback. As requested, we provide additional experiment results on Fashion-MNIST (see Table 3 of our revised submission), where we observe that CLIMB outperforms its competitors in both highly heterogeneous and moderate heterogeneous settings. In our future revision, we will include the experiment on Fashion-MNIST under more hyperparameter configurations.

---

### Official Review · Reviewer_u6Lr · 2021-11-02

**Correctness:** 3
**Technical Novelty And Significance:** 3
**Empirical Novelty And Significance:** 2
**Recommendation:** 6
**Confidence:** 3

**Main Review:**

Overall, the paper is well written, with sufficient background and motivating examples on the problem under consideration. There exists some minor grammatical issues in Section 2 which needs to be carefully reviewed.
The empirical results clearly shows benefits of the solution setup. However, there are 3 suggestions to improve the paper: (1) Use of more complex datasets, (2) Comparison with other FL algorithms mentioned in the paper, even under additional settings such as active learning, MAB etc. (3) Need for ablation study on alpha.

Though MNIST and CIFAR10 is typically used as a standard dataset in literature, there exists more complex image datasets where the proposed method may struggle. From the current description, the limitations of the methods are not clear. Where does the method not work? If it works, why does it work? Here, comparison with existing techniques may aid in showing reasons for enhanced model performance. Finally, it would be interesting to see how variation in alpha affects model performance. If alpha=0.5, would the model still work relative to its baseline? When deploying the model in practice, alpha may not be known.

It is also not clear on how training and deployment of the algorithm work in practice. If a new client is added to the federated system, how does that affect model performance when shift in global data imbalance change? It would be good to discuss future work and how its applicability to multiclass classification would affect computational performance.

**Summary Of The Paper:**

The paper describes an agnostic constraint learning method for handling class imbalance in a federated learning setting. Here, a client may have a minority class which is globally (across clients) a majority class. By setting up the constraint learning using Lagrange multipliers, the theoretical solution and convergence guarantees is shown, along with empirical evidence of the methods superiority.

**Summary Of The Review:**

The paper provides good theoretical and empirical justification of a unique problem of addressing extreme local imbalance in a federated data system while maintaining data privacy. However, additional supporting material and stronger empirical support may be required to justify strong conclusions in the paper.

---

> ### Author Response · Authors · 2021-11-23
> **Response to Reviewer u6Lr**
>
> We thank the reviewer for the insightful review. We have taken all the points into consideration and made the modifications accordingly.
>
> > Use of more complex datasets
>
> We agree with the reviewer that more complex dataset should be explored. Due to time limitation, we only provide additional results on the Fashion-MNIST dataset (see Table 3 of our revised submission). Simulations show that our method attain a better performance on this dataset as well. We will work on more datasets like EMNIST and CIFAR100 in our future revision.
>
> > Comparison with other FL algorithms mentioned in the paper, even under additional settings such as active learning, MAB etc
>
> We agree with the reviewer that we can further improve the quality of our paper by comparing with more baselines. However, due to time limitation, we defer this task to our future revision.
>
> > Need for ablation study on alpha
>
> We agree with the reviewer that different configurations should be taken into consideration and we should clearly state the limitation of our method. We hence perform an ablation study on $\alpha$ by ranging $\alpha$ from $0.0$ to $0.5$ (see Table 4 of our revised submission). We observe that CLIMB has advantage over the included methods up till $\alpha=0.4$ and the advantage is reduced as $\alpha$ grows larger. For $\alpha=0.5$, Ratio-Loss outperforms CLIMB.
>
> > It is also not clear on how training and deployment of the algorithm work in practice.
>
> We thank the reviewer for pointing this out since this discussion will help improve the quality of our paper. The deployment of the classifier does not require the use of constraints. That is to say, once the classifier has converged, we can deploy it in a real world scenario, we do not require the use of any dual variable for the deployment of the learn classifier. We are not sure what the review means about "how training of the algorithm work in practice". Could you be more specific?
>
> > If a new client is added to the federated system, how does that affect model performance when shift in global data imbalance change? It would be good to discuss future work and how its applicability to multiclass classification would affect computational performance.
>
> This an excellent point raised by the reviewer. Adding new clients to the system is a common event in practical federated learning. Suppose that CLIMB has already converged and we have obtained a trained model. When a new client joins and the global data imbalance ratio changes, a simple strategy would be to use the current trained model as warm start (the corresponding dual variable should also be used) and then run CLIMB for a few more rounds (dual variable the new client should be initialized as zero). We conjecture that if the new client contains a lot of minority data, the model performance can be significantly enhanced. We also believe that we can borrow ideas from the literature of continual learning to facilitate the procedure when new clients join the training. This would be an interesting future work.
>
> In terms of the computational cost of CLIMB, since the primal step of CLIMB solves a reweighted version of the standard FL objective, its computational cost is the same as standard FL solvers. In the dual step, every client simply performs a one dimensional scalar update which is negligible computationally.

---

### Official Review · Reviewer_8nbc · 2021-11-03

**Correctness:** 2
**Technical Novelty And Significance:** 3
**Empirical Novelty And Significance:** 3
**Recommendation:** 6
**Confidence:** 4

**Main Review:**

Strengths:
1. Their method is agnostic to class distribution of the client data, which satisfies the privacy requirements of FL.

2. There is no need for active client selection in their method which is more practical than other similar work.

Weaknesses:
1. My first concern is about the Assumption 3.4. It seems too strong to me. Can you provide more FL literature which support this assumptions? I also hope you could provide a simple example which satisfies this assumption.

2. I am quite confused with the Theorem 3.2. What is the relationship between Theorem 3.2 and convergence of FL algorithms? Maybe you could combine it with Theorem 3.1 and add more detailed explanations to them.

3. The settings of  the experiments is ambiguous. In the caption of Table 1&2, what does “sufficiently many communication rounds” means?
I recommend the authors should add some “Accuracy and Epoch” figures to show how fast the accuracy increases in their method and other baselines.

4. The results in Table 2 show that their method can achieve a better performance than Table 1. I think the authors should provide some theoretical analysis to explain this phenomenon.

5. In the section of introduction, the authors mention that a drawback of Yang et al., 2020 is that “a subset of the client’s local data need to be shared”. Why?

6. Minor typos. “they will a higher probability to”->"there will ..."


**Summary Of The Paper:**

The authors design a method, CLIMB, to solve the severe class imbalance issues in FL problem. In their method, they propose a constrained FL formulation (CFL) and adopt the method of Lagrange multipliers to solve this problem. They present some theoretical and experimental results to prove that their method is effective.

**Summary Of The Review:**

Although reducing the class imbalance in FL is an interesting topic and the method in this paper is new, I still think their method are not well-supported by theories and experiments according to the above discussion. Therefore, I think it is marginally below the acceptance threshold and needs further improvements.

---

> ### Author Response · Authors · 2021-11-23
> **Response to Reviewer 8nbc**
>
> > My first concern is about the Assumption 3.4 ... I also hope you could provide a simple example which satisfies this assumption.
>
> This is an excellent point raised by the reviewer. We acknowledge that for general nonconvex constraints, Assumption 3.4 is too strong. However, we emphasize that the Lipschitz continuity constant of Assumption 3.4 can be explicitly calculated as the optimal dual variable for the problem of interest in our paper when the cross entropy is used as loss function (see more discuss in **Appendix C.2**). As a result, we remove Assumption 3.4 in our revision and modify Theorem 3.1 and its proof accordingly.
>
> > I am quite confused with the Theorem 3.2... Maybe you could combine it with Theorem 3.1 and add more detailed explanations to them.
>
> Theorem 3.2 provides a convergence guarantee for the dual variable in the last iteration, which can be regarded as a proxy of the convergence of the proposed primal-dual method (CLIMB in Algorithm 1). We apologize for the confusion caused by our presentation and we agree with the reviewer that it is important to provide a result that is comparable to the other FL algorithms. However, we also would like to mention that this comparison is not easy for the following reasons.
>
> 1. The problem of interest in our paper is has a non-convex objective function with a non-convex constraint set, while previous FL algorithms only consider unconstrained (non-)convex problems.
> 2. Our proof works under the oracle Assumption 3.4 which is not required by previous FL algorithms.
>
> While we cannot provide a result that is exactly comparable with existing FL algorithms, to relate our work to existing arts, we present another type of guarantee for the convergence of CLIMB from a primal perspective (Theorem 3.2 is from a dual perspective): in Theorem C.1 of our revised submission, we show that
>
> 1. (optimality) in an ergodic sense, the objective function converges to a ball of the global optimal whose radius depends on the dual step size,
> 2. (feasibility) in an ergodic sense, the iterates are feasible, i.e. the average constraint violation diminishes to zero as the number of iterations $T$ goes to infinity.
>
> While it would be ideal to improve the above ergodic result to the guarantee for the last iterate (just like Theorem 3.2 for the dual variable), ergodic type convergence is common for non-convex problems, e.g. for an unconstrained non-convex minimization problem, the standard convergence result is that the norm of the gradient diminishes to zero as the number of iterations goes to infinity in an ergodic sense (see for example Theorem 1 of (Acar et al., 2020)).
>
> > The settings of the experiments is ambiguous ... to show how fast the accuracy increases in their method and other baselines.
>
> In the first paragraph of Section 4.1, we discussed the concept of “sufficiently many communication rounds”. Specifically, 5000 rounds for CIFAR10 and 1000 rounds for MNIST.
>
> We agree with the reviewer and we have added the plots that show accuracy per epoch in Figure 2 of our revised submission. Due to the time limitation, we provide results on CIFAR10 with the heterogeneity $\alpha$ being $0.1$ and $0.2$ only. We will add the rest of the plots in our later revision. We thank the reviewer for pointing this out.
>
> > The results in Table 2 show that their method ... provide some theoretical analysis to explain this phenomenon.
>
> We apologize for the confusion. In Table 1, we consider the massive device setting where there are totally 500 clients while in Tabl 2, we consider the moderate device setting where there are totally 100 clients. The reviewer correctly points out that the performance deteriorates when more clients are involved in the training (hence less local data). This is a pattern that can be commonly seen in Federated Learning (see for example Tabl 1 of (Acar et al., 2020)). This phenomena can be explained by the fact that adding more clients translates into less samples per client, which generates clients that are more heterogeneous and adds a larger bias in each particular gradient.
>
> > In the section of introduction, the authors mention that a drawback ... Why?
>
> The reviewer asks for a clarification in regards to the paper Yang et al., 2020, to which we said that "a subset of the client’s local data need to be shared". We agree with the reviewer that we may have been unspecific here. In Yang et al., the clients do not share portions of the dataset. However, what the clients share is the gradient per class, which is a proxy for the relative number of samples per class (see Theorem 1 Yang et al.). Thus, clients do share data regarding their local distribution, which attempts against the privacy concern of federated learning. We have corrected the text in the virtue of clarity.
>
> > Minor typos. “they will a higher probability to”->"there will ..."
>
> We thank the reviewer for pointing out this typo and it is fixed in our revision.

---

### Official Review · Reviewer_obo5 · 2021-11-03

**Correctness:** 3
**Technical Novelty And Significance:** 3
**Empirical Novelty And Significance:** 3
**Recommendation:** 6
**Confidence:** 3

**Main Review:**

My main concern about the paper is the presentation of CLIMB as the method counteracting the issue of class imbalance. The method itself directly optimizes a problem that imposes small differences in classification performance among the clients. This certainly improves the fairness of the trained model, but it does not help when data is imbalanced. The authors demonstrate that CLIMB improves performance on minority classes in the imbalanced setting; however, it seems that it is only because the authors distribute the data across the clients in a special manner. Concretely, the dataset of selected clients contains a lot of minority examples, to the extent that locally the minority class has much more examples than majority classes. Therefore, a method that looks for a fair model (similar performance for each client) naturally also improves for imbalanced classes. However, it is difficult to say to what extent such particular data distribution occurs in practice. Moreover, the approach seems not to handle class imbalance in any way if the data is not distributed in such a particular way.

The second concern is about privacy. The approach seems to require sending lambda values for each client to the server, which are directly related to the difference between average performance among the clients and the client's performance. Therefore, by observing the value of lambda, one can deduce if the performance on client's data is good. Since the classifier usually better recognizes majority classes (particularly during training), the high lambda value possibly indicates clients with many minority class data? Have the authors considered this issue?



**Summary Of The Paper:**

The paper proposes an interesting new federated learning approach called CLIMB that leverages a special formulation of the FL problem as optimization with constraints. The approach aims to improve the classification performance of FL method on class imbalanced data.


**Summary Of The Review:**

In general, the paper is well written, and the algorithm presentation is clear. The experimental evaluation is, in my opinion, insufficient if we interpret CLIMB as an imbalanced learning method (see the second paragraph of the review). For instance, more different ways of distributing data among the clients should be investigated.

---

> ### Author Response · Authors · 2021-11-23
> **Response to Reviewer obo5**
>
> > My main concern about the paper is the presentation of CLIMB as the method counteracting the issue of class imbalance .... This certainly improves the fairness of the trained model, but it does not help when data is imbalanced.
>
> We cannot agree with the reviewer about this comment. As acknowledged by the reviewer, through our extensive empirical study, we have shown that at least under the type of heterogeneity considered in our paper, our approach is able to address the class imbalance problem. In fact, our method consistently outperforms previous arts by a large margin. We also emphasize that such a heterogeneity generating process is a common practice in the literature of federated learning, see e.g. (Karimireddy et al., 2020) and (Hsu et al., 2019).
>
> > The authors demonstrate that CLIMB improves performance on minority classes in the imbalanced setting ... Moreover, the approach seems not to handle class imbalance in any way if the data is not distributed in such a particular way.
>
> To showcase that our approach has a broader applicability beyond the type of heterogeneity considered in our paper, we conduct additional experiments on the Dirichlet type heterogeneity which is another commonly used heterogeneity generating process, see e.g. (Hsu et al., 2019) and (Acar et al., 2020). The results on the CIFAR10 and Fashion-MNIST datasets are reported in **Table 3 in Section B** of the appendix in our revised submission. We can observe that CLIMB has clear advantage over the previous arts in both highly heterogeneous setting (Dirichlet setting with hyperparameter 0.3) and moderate heterogeneous setting (Dirichlet setting with hyperparameter 10.0). In all configurations, we are still able to observe the advantage of CLIMB over the included baselines.
>
> > The second concern is about privacy. The approach seems to require sending lambda values for each client to the server, which are directly related to the difference between average performance among the clients and the client's performance. Therefore, by observing the value of lambda, one can deduce if the performance on client's data is good. Since the classifier usually better recognizes majority classes (particularly during training), the high lambda value possibly indicates clients with many minority class data? Have the authors considered this issue?
>
> We thank the reviewer for pointing out this important problem. We acknowledge that a client corresponding to a large dual variable is more likely to possess minority data. However, we emphasize that under the standard federated learning paradigm, CLIMB can be implemented in a privacy-preserving manner: one **cannot** recover the dual variables on the server side.
>
> To achieve this goal, first observe that the weight computing step and dual update step of CLIMB (line 4 and line 6 of algorithm 1 respectively) can be carried out locally as long as the client has access to the global average dual variable $\bar \lambda$ and the global average loss $\bar f(\theta^{t+1})$ since other terms only involve local information. In order to compute these two terms on the server in a privacy-preserving manner, the problem can be simplified as follows:\
> Assuming that each client privately holds a quantity $a_i$, how can we compute the global average $\bar a = \frac{1}{n}\sum_{i\in[n]} a_i$ without revealing the quantities $a_i$'s to the server?\
> Assuming there is a secret key that can be used to encrypt the quantity $a_i$ is available to the clients but **not** to the server, then with the **Homomorphic Encryption** technique the clients can 1. encrypt $a_i$ locally, 2. send it to the server to compute homomorphically (the server **cannot** decrypt the individual contribution $a_i$ since it does not have the key), and 3. decrypt the received average locally using the secret key. In this way, both the average dual variable and the global average loss can be computed without being revealed to the server. The aforementioned scheme is often known as secure multi-party computation and the secret key is usually implemented through a trusted 3rd party (for example see the discussion on page 42 of the monograph [1]). We also note that while Homomorphic encryption can be computational expensive, since we are encrypting two scalars, the cost is negligible.
>
> [1] Advances and Open Problems in Federated Learning. https://arxiv.org/abs/1912.04977

---

### Decision · Program_Chairs · 2022-01-20

**Decision:**

Accept (Poster)

**Comment:**

This paper presents a method to handle class imbalance in federated learning, while accounting for data heterogeneity and privacy. The key idea is to solve a constrained optimization problem where the difference between the global and local objective values has to be less than some parameter $\epsilon$. The paper proposes a primal-dual optimization algorithm called CLIMB to solve this constrained FL problem. The paper presents a theoretical analysis of the algorithm, as well as experimental results.

All the reviewers found the formulation interesting and novel and gave a positive assessment of the paper. Reviewer obo5 had some concerns about whether the optimization problem is improving fairness and getting reduced class imbalance as a side benefit or whether it is directly addressing class imbalance. After discussion with the authors, their concerns were partially addressed. Reviewer 8nbc had concerns about the assumptions and theoretical analysis. Their concerns were also mostly addressed by the authors during the discussion phase. I suggest the authors to also address Reviewer u6Lr and Reviewer Mp3G's concerns about experimental results and citing related work respectively when they revise the paper.

Overall, I recommend acceptance of the paper, and strongly encourage the authors to take the reviewers' suggestions about 1) fairness connections, 2) privacy connections, 3) theoretical analysis, 4) experimental results, and 5) prior work into account when revising it.